# Patient and healthcare provider knowledge, attitudes and barriers to handover and healthcare communication during chronic disease inpatient care in India: a qualitative exploratory study

Claire Humphries [1] , Suganthi Jaganathan,[2,3] Jeemon Panniyammakal,[2,3,4] Sanjeev K Singh,[5] Shifalika Goenka,[2,3] Prabhakaran Dorairaj,[2,3] Paramjit Gill,[6] Sheila Greenfield,[1] Richard Lilford,[7] Semira Manaseki-Holland[1]

CH and SJ contributed equally.

For numbered affiliations see end of article.

**Correspondence to**
Dr Semira Manaseki-Holland;
s.manasekiholland@bham.ac.uk

## ABSTRACT

**Objectives** 1) To investigate patient and healthcare provider (HCP) knowledge, attitudes and barriers to handover and healthcare communication during inpatient care. 2) To explore potential interventions for improving the storage and transfer of healthcare information.

**Design** Qualitative study comprising 41 semi-structured, individual interviews and a thematic analysis using the Framework Method with analyst triangulation.

**Setting** Three public hospitals in Himachal Pradesh and Kerala, India.

**Participants** Participants included 20 male (n=10) and female (n=10) patients with chronic non-communicable disease (NCD) and 21 male (n=15) and female (n=6) HCPs. Purposive sampling was used to identify patients with chronic NCDs (cardiovascular disease, chronic respiratory disease, diabetes or hypertension) and HCPs.

**Results** Patient themes were (1) *public healthcare service characteristics*, (2) *HCP to patient communication* and (3) *attitudes regarding medical information*. HCP themes were (1) *system factors*, (2) *information exchange practices* and (3) *quality improvement strategies*. Both patients and HCPs recognised public healthcare constraints that increased pressure on hospitals and subsequently limited consultation times. Systemic issues reported by HCPs were a lack of formal handover systems, training and accessible hospital-based records. Healthcare management communication during admission was inconsistent and lacked patient-centredness, evidenced by varying reports of patient information received and some dissatisfaction with lifestyle advice. HCPs reported that the duty of writing discharge notes was passed from senior doctors to interns or nurses during busy periods. A nurse reported providing predominantly verbal discharge instructions to patients. Patient-held medical documents facilitated information exchange between HCPs, but doctors reported that they were not always transported. HCPs and patients expressed positive views towards the idea of introducing patient-held booklets to improve the organisation and transfer of medical documents.

### Strengths and limitations of this study

► This is the first qualitative study, as far as the authors are aware, to explore both patient and healthcare provider knowledge, attitudes and barriers to multiple areas of handover and healthcare communication for inpatients with chronic disease in India.

► The number of interviews from both patients and healthcare providers facilitated data saturation and provided a range of significant perspectives.

► Analyst triangulation corroborated data analysis and strengthened the credibility of the study.

► The accuracy of recall of patients interviewed at home (ie, following hospital discharge) may have been limited by the delay between study recruitment and subsequent data collection.

► Awareness of the interviewer's context as a public health researcher may have resulted in participants distorting their responses to minimise critical judgement.

**Conclusions** Handover and healthcare communication during chronic NCD inpatient care is currently suboptimal. Structured information exchange systems and HCP training are required to improve continuity and safety of care during critical transitions such as referral and discharge. Our findings suggest that patient-held booklets may also assist in enhancing handover and patient-centred practices.

## INTRODUCTION

The increasing burden of chronic, non-communicable diseases (NCDs), such as cardiovascular disease, diabetes and chronic respiratory disease, has become a global pandemic that is disproportionately affecting low-income and middle-income

countries (LMICs).[1] This is placing great demand on under-resourced health systems that can only be relieved by employing efficient and integrated approaches to healthcare management. Central to efficiency and integration in healthcare is effective handover communication, which involves the exchange of patient-specific information between healthcare providers (HCPs) and between HCPs and patients/carers to ensure continuity and safety of care.[2]

Between HCPs, information exchange is critical during clinical handovers, which are the points in care where information, responsibility and accountability for patient care are transferred from one HCP to another.[3] This is because safe and effective treatment can only be maintained if all relevant information has been shared and understood.[4] A wealth of research from high-income countries (HICs) has evidenced the association between communicative breakdowns during care transitions and risks to patient safety. These risks are pervasive throughout inpatient care and include delays in diagnosis, medication errors and life-threatening adverse events.[5][6] In addition, one in five patients experience adverse events following hospital discharge, and research has established a link between such events and deficient handover communication.[7–10] Between HCPs and patients, effective information exchange is also vital as patients can provide valuable information to those involved at various stages of their care pathway.[4] Excellent HCP–patient healthcare communication further empowers patients to become active participants in their healthcare management; this is a key aspect of patient-centred care which has been linked to improved patient satisfaction and outcomes.[11]

Despite the established importance of handover communication for health systems functioning and patient safety in HICs, there has been a relative dearth of LMIC-based research focusing on this topic.[12] A recent (2019) study from South Africa has found inadequate discharge planning to be a significant contributor to avoidable causes of hospital readmission.[13] Across India, a handful of predominantly single-site studies have evaluated and described deficiencies in information exchange during referrals, hospital shift change and discharge.[14–19] The current study forms part of a series completed for a project investigating handover and continuity of care for patients with chronic NCDs in the states of Kerala and Himachal Pradesh in India. The first study to have been disseminated focused on outpatient care, which found issues such as suboptimal recording of information within patient-held medical documents and a lack of formal information exchange systems between levels of care.[20]

Given these emerging challenges and the established link between deficient handover communication during inpatient care and risks to patient safety, the current study was conducted to gain novel insight into healthcare information transfer during chronic NCD inpatient care across the same study areas of India. The primary objective was to explore knowledge, attitudes and barriers to handover and healthcare communication during the following points of inpatient care: (1) referral/transfer (ie, communication between HCPs and between HCPs and patients when referring and/or transferring patients) and (2) hospital admission and discharge (ie, communication between HCPs and patients regarding condition, treatment and/or management during hospital admission and discharge). A secondary objective was to explore potential interventions to improve the storage and transfer of key healthcare information.

## METHODS

### Overview

We report findings from a qualitative study of handover and healthcare communication for inpatients with chronic NCDs in two Indian states. This study was conducted from December 2014 to November 2015 across three public hospitals: one rural secondary-care hospital in the state of Himachal Pradesh, and one periurban secondary-care and one urban tertiary-care hospital in the state of Kerala. These settings were selected to capture a range of hospital types within different geographical settings. We selected public rather than private facilities as these are where a large proportion of socioeconomically vulnerable patients access healthcare. See online supplementary files S1 and S2 for further information regarding the Indian healthcare system and study settings.

### Participant recruitment

#### Patients

Patients were recruited opportunistically from hospitals by trained research assistants (n=6).[21] Purposive sampling was used to identify individuals who met the following inclusion criteria[22]: adults (18+ years) admitted to hospital within 24 hours of a researcher first meeting them due to complications from one of the following chronic NCDs: diabetes mellitus, cardiovascular disease, chronic respiratory disease or hypertension. The identification process took place via researchers approaching ward nurses and asking them about patient demographics and admission details; patients were excluded if judged too unwell to participate by ward nurses. Patients who met the inclusion criteria were provided with verbal and documented study information. Written consent was obtained from literate patients. For illiterate patients, oral consent was obtained along with a thumbprint and signature from a literate witness (ie, family member/carer) in line with WHO guidelines.[23] Patients were recruited until theoretical saturation was achieved.[24] A total of 20 patients participated.

#### Healthcare professionals

HCPs were recruited from study hospitals by trained research assistants (n=6). Due to the busy nature of the study settings, opportunistic sampling was used to recruit as many HCPs as possible with a range of roles and experience.[21] During recruitment, if HCPs stated they were too busy to answer questions, they were marked

as 'unavailable' and not approached again that day; this did not exclude them from participating at another time. HCPs were also recruited until theoretical saturation was achieved.[24] A total of 21 HCPs participated.

## Sample size

As well as saturation being reached for both participant groups independently, the resulting sample size of 41 participants for this study was in accordance with Baker et al's[25] review of sample sizes used in qualitative literature, indicating it was sufficient for achieving overall data saturation.

## Data collection

The inpatient data analysed for this study are independent from the outpatient study and were collected from different patients using separate topic guides. Regarding HCP data, this study involves secondary analysis of HCP interviews (n=17) included in the outpatient study from participants who were also involved in inpatient care. A small number of additional interviews with HCPs solely involved in inpatient care (n=4) have also been analysed in this study. All HCP interviews in the India handover project were conducted within the same study period and used the same topic guide (as most HCPs in the study areas worked with both outpatients and inpatients on a daily basis).

All interview data were collected entirely by the lead Indian researcher (SJ, an experienced public health researcher), who was familiar with the study areas and fluent in alllanguages used during interviews. Full consideration was given prior to and throughout data collection to ensure that SJ was aware of the potential limitations of working with participants from culturally and linguistically diverse backgrounds. SJ was not involved in the treatment of patients or previously known to HCPs.

The majority (n=16) of patient interviews took place in study hospitals. Due to a lack of private spaces, interviews were conducted on inpatient wards in as quiet and private a manner as possible. All participants consented to this and it was ensured that HCPs were not present during patient interviews. In addition, a small number of patient interviews (n=4) took place in patients' homes either 5 weeks (n=2) or 4 months (n=2) following hospital discharge, as this was more convenient for them (ie, during recruitment they were in the process of being discharged and leaving hospital). The specific follow-up times coincided with community visits being completed for another quantitative study within the India handover project, which the four patients were also participating in. All HCP interviews took place in hospital offices. Interviews with patients and HCPs were conducted in either English, Hindi, Malayalam or a mixture, depending on interviewee preference, and audio-recorded using a digital Dictaphone.

Data collection took place in two stages. In the first stage (December 2014–October 2015), preprepared topic guides were used to guide interviews. These were developed using relevant handover literature and local knowledge of health systems functioning within the study areas. They were also piloted over three rounds prior to commencement of data collection to ensure they were clear, as well as culturally and contextually appropriate. Patient topic guides included open-ended questions focusing on healthcare utilisation and experiences and attitudes of healthcare visits and information exchange. The HCP topic guides differed slightly to capture information on health systems policies and/or practices; they also included questions regarding handover training and potential strategies for improving practices.

Following the first stage of data collection, on 11 October 2015, a handover expert meeting took place in Delhi, India to present preliminary findings from the India handover project and discuss possible interventions. Researchers from the University of Birmingham and the University of Warwick (UK) facilitated the presentation of results and group discussions at the meeting. Representatives (n=27) from the following international, Indian national and state-level organisations participated: WHO, The World Bank, ACCESS Health International, the Ministry of Health and Family Welfare, the Public Health Foundation of India, the National Centre for Disease Control, the Centre for Chronic Disease Control, the National Health Systems Resource Centre, the All India Institute of Medical Sciences, Aga Khan Health Services, Amrita Institute of Medical Science, and Fortis Hospitals. During discussions, a consensus was reached that patient-held booklets were likely to be an acceptable and sustainable intervention to improve information exchange. This was based on the international success of similar patient-held records used in maternal healthcare around the world.[26–30] It also took into account the delays in developing universal electronic information systems and the fact that such systems will not necessarily address the quality of communication between HCPs and patients. Overall, it was opted as the most pragmatic, cost-effective intervention. Multiple experts also felt that booklets could improve patient self-management if they contained disease-specific advice.

Therefore, following the meeting, the second stage of qualitative data collection (October–November 2015) commenced. Topic guides were updated to include questions regarding the utility of patient-held booklets. In addition, if participants stated they had limited time, researchers interviewed them using a shortened topic guide containing targeted questions on patient-held booklets and medical documents.

## Data analysis

All audio recordings of interviews were transcribed verbatim and, if necessary, translated into English by SJ. All translations were crosschecked for accuracy by a qualitative expert in India (SGo, professor of bioethics and social and behavioural sciences with expertise in NCDs), who was also familiar with the study settings and fluent in all languages used during interviews. Following this,

the transcripts were sent to the lead UK researcher (CH, public health PhD student) for analysis. CH became familiar with the study settings prior to analysis during multiple research-related site visits that were facilitated by the Public Health Foundation of India and the Ministry of Health and Family Welfare in Kerala.

Data were analysed using the Framework Method,[31] as this is the method most commonly used for semi-structured interview transcripts. An inductive thematic approach to analysis used in grounded theory was employed,[32 33] which focused on analysing interviews in their entirety and identifying concepts relevant to handover and healthcare communication during inpatient care that emerged from interviews. Analysis occurred through the following stages central to the Framework Method: transcription, familiarisation, coding, charting and interpretation. Over a 1-month period, familiarisation with the data took place via the slow reading of transcripts, and CH consulted with SJ to gain a clear understanding of interview contexts. Once this was complete, coding began and two transcripts were chosen at random from each batch of interviews (ie, two patient and two HCP transcripts) for independent coding by an additional UK analyst (SGr, professor of medical sociology with expertise in cross-cultural research) for analyst triangulation.[34] Patient and HCP transcripts were coded separately in order to be able to assess similarities and differences between participant groups; patient transcripts were coded first. The coding process involved further familiarisation with the data, followed by open coding where certain transcript content was highlighted and allocated descriptive labels (codes) to interpret the phenomena identified in the text. The development of codes and themes was entirely data-led and analysed manually.[35]

Microsoft Excel was used to organise participant codes. CH created initial categories by clustering similar codes developed from the two randomly selected patient and HCP transcripts. CH and the additional UK analyst (SGr) then met to discuss their analyses. As both had produced similar codes and concepts, the categories that were created were mutually agreed on. CH then continued with category development until all transcripts had been coded and inserted into the spreadsheet. Following analysis of 20 patient and 21 HCP transcripts, no new categories had been produced. This served as confirmation that data saturation had been met.[24]

Following coding, categories were grouped into subcategories and linked to produce themes. Then, via the process of charting,[31 35] themes for each participant group were used to create a framework matrix into which participants' quotes were inserted, corresponding to their representative subcategory. This provided a visual representation of themes, which facilitated the mapping and interpretation of the data. After completing separate analysis of patient and HCP data, the results of both participant groups were compared to assess similarities and differences between their reports of knowledge, attitudes and barriers to handover and healthcare communication.

A Venn diagram was used to summarise the separate and overlapping content, which was linked to subcategories from the original themes.

### Patient and public involvement

Patients and the public were not involved in the initial design of this study. Patients and carers were first involved during the pilot phase prior to formal data collection, where the topic guides, consent and study information sheets were piloted over three rounds. During this time, they were consulted and given the opportunity to provide feedback to ensure the study materials were clear and culturally and contextually appropriate. Patients and the public were not involved in any other aspect of the study recruitment or conduct, but findings have been disseminated publicly via an expert meeting (including professionals working with patient groups) and open access web pages.

## RESULTS
### Patient characteristics

Twenty male (n=10) and female (n=10) patients aged between 25 and 72 years old were interviewed. Participants' background characteristics varied (table 1). Patients completed interviews in English (n=11), Hindi (n=4), Malayalam (n=4) and a mixture of Hindi and English (n=1).

### Healthcare professional characteristics

Twenty-one male (n=15) and female (n=6) HCPs aged between 22 and 55 years old were interviewed. HCP roles included doctors (n=17), nurses (n=2), a pharmacist (n=1) and a medical records officer (n=1). HCP qualifications and experience varied (table 2). HCPs completed interviews in English (n=15), Hindi (n=2), Malayalam (n=2) and a mixture of Hindi and English (n=2).

### Charted data

During analysis of patient and HCP data, three themes (with subcategories) emerged for each participant group. Patient themes were (1) public healthcare service characteristics, (2) HCP to patient communication and (3) attitudes regarding medical information (table 3). HCP themes were (1) system factors, (2) information exchange practices and (3) quality improvement strategies (table 4).

Following separate analysis of patient and HCP data, the results of both participant groups were compared to assess similarities and differences between their reports of knowledge, attitudes and barriers to handover and healthcare communication; the results of this comparison are displayed in figure 1. The similarities will be described first, followed by the differences. To ensure confidentiality, numerical pseudonyms have been used when presenting quotes.

### Overlapping content
#### Public healthcare constraints

During interviews, a number of patients reported that they chose to visit public hospitals because of the better

**Table 1** Patient characteristics

| Characteristics | 1 | 2 | 3 | 4 | 5 | 6 | 7 | 8 | 9 | 10 | 11 | 12 | 13 | 14 | 15 | 16 | 17 | 18 | 19 | 20 | n (%) |
|---|---|---|---|---|---|---|---|---|---|---|---|---|---|---|---|---|---|---|---|---|---|
| Age | 65 | 45 | 70 | 58 | 71 | 56 | 57 | 70 | 55 | 25 | 72 | 50 | 55 | 69 | 70 | 50 | 70 | 70 | 70 | 70 | 25–72 |
| Sex | | | | | | | | | | | | | | | | | | | | | |
| Male | ✓ | ✓ | ✓ | ✓ | ✓ | | | | | ✓ | ✓ | ✓ | | | ✓ | ✓ | | | | | 10 (50.0) |
| Female | | | | | | ✓ | ✓ | ✓ | ✓ | | | | ✓ | ✓ | | | ✓ | ✓ | ✓ | ✓ | 10 (50.0) |
| Literacy | | | | | | | | | | | | | | | | | | | | | |
| Illiterate | | | | | | | | ✓ | | | | | ✓ | | ✓ | ✓ | ✓ | ✓ | ✓ | ✓ | 8 (40.0) |
| Literate | ✓ | ✓ | ✓ | ✓ | ✓ | ✓ | ✓ | | ✓ | ✓ | ✓ | ✓ | | ✓ | | | | | | | 12 (60.0) |
| Education level | | | | | | | | | | | | | | | | | | | | | |
| None/minimal primary school level | | | | | | | | ✓ | | | | ✓ | ✓ | | ✓ | ✓ | ✓ | ✓ | ✓ | ✓ | 9 (45.0) |
| Completed lower primary school | | | | | ✓ | | ✓ | | | | | | | | | | | | | | 2 (10.0) |
| Completed upper primary school | | | | | | ✓ | | | | | | | | | | | | | | | 1 (5.0) |
| Completed secondary school | | | | | | | | | ✓ | | | | | | | | | | | | 1 (5.0) |
| University graduate (or above) | ✓ | | | ✓ | | | | | | ✓ | ✓ | | | | | | | | | | 4 (20.0) |
| No data | | ✓ | ✓ | | | | | | | | | | | ✓ | | | | | | | 3 (15.0) |
| Employment status | | | | | | | | | | | | | | | | | | | | | |
| Employed | | ✓ | ✓ | ✓ | ✓ | ✓ | ✓ | | | | | ✓ | ✓ | | | | | | | | 8 (40.0) |
| Unemployed | | | | | | | | ✓ | ✓ | | ✓ | | | | ✓ | ✓ | ✓ | ✓ | | ✓ | 8 (40.0) |
| Student | | | | | | | | | | ✓ | | | | | | | | | | | 1 (5.0) |
| Retired | ✓ | | | | | | | | | | | | | ✓ | | | | | ✓ | | 3 (15.0) |
| Chronic NCD(s) (related to admission)* | | | | | | | | | | | | | | | | | | | | | |
| Chronic respiratory disease | ✓ | | | | | | | ✓ | ✓ | | | | ✓ | | | ✓ | | | ✓ | | 6 (30.0) |
| Diabetes | ✓ | | | | | | | | | | | | ✓ | ✓ | ✓ | ✓ | ✓ | ✓ | | ✓ | 8 (40.0) |
| Hypertension | | | | | | ✓ | | | | | ✓ | ✓ | | | ✓ | | ✓ | ✓ | | | 6 (30.0) |
| Cardiovascular disease (other than hypertension alone) | ✓ | ✓ | | ✓ | | ✓ | | | ✓ | ✓ | ✓ | ✓ | | | | | ✓ | | | | 9 (45.0) |
| Language(s) used during interview | | | | | | | | | | | | | | | | | | | | | |
| English (only) | ✓ | ✓ | ✓ | ✓ | | | | | | | | | | ✓ | ✓ | ✓ | ✓ | ✓ | ✓ | ✓ | 11 (55.0) |
| Hindi (only) | | | | | | | | | | ✓ | ✓ | ✓ | ✓ | | | | | | | | 4 (20.0) |
| Malayalam (only) | | | | | | ✓ | ✓ | ✓ | ✓ | | | | | | | | | | | | 4 (20.0) |
| English and Hindi (mixture) | | | | | ✓ | | | | | | | | | | | | | | | | 1 (5.0) |

*Patients could select more than one answer to this question.
NCD, non-communicable disease.

**Table 2** Healthcare professional characteristics

| Characteristics | 1 | 2 | 3 | 4 | 5 | 6 | 7 | 8 | 9 | 10 | 11 | 12 | 13 | 14 | 15 | 16 | 17 | 18 | 19 | 20 | 21 | n (%) |
|---|---|---|---|---|---|---|---|---|---|---|---|---|---|---|---|---|---|---|---|---|---|---|
| Age | 44 | 24 | 33 | 25 | 23 | 39 | 44 | 35 | 52 | 50 | 50 | 43 | 50 | 40 | 46 | 55 | 22 | 35 | 35 | 45 | 35 | 22–55 |
| Sex | | | | | | | | | | | | | | | | | | | | | | |
| Male | ✓ | | ✓ | ✓ | ✓ | | ✓ | | ✓ | ✓ | ✓ | ✓ | ✓ | ✓ | ✓ | ✓ | ✓ | | | ✓ | | 15 (71.4) |
| Female | | ✓ | | | | ✓ | | ✓ | | | | | | | | | | ✓ | ✓ | | ✓ | 6 (28.6) |
| Qualification/s* | | | | | | | | | | | | | | | | | | | | | | |
| Doctor of Medicine (MD) | ✓ | | ✓ | | | | ✓ | ✓ | ✓ | ✓ | ✓ | ✓ | ✓ | ✓ | ✓ | ✓ | ✓ | | | | | 13 (61.9) |
| Master of Public Health (MPH) | | | | | | ✓ | | | | | | | | | | | | | | | | 1 (4.8) |
| Bachelor of Medicine (MBBS) | ✓ | ✓ | ✓ | ✓ | ✓ | ✓ | ✓ | ✓ | ✓ | ✓ | ✓ | ✓ | ✓ | ✓ | ✓ | ✓ | ✓ | | | | | 17 (81.0) |
| BSc Nursing | | | | | | | | | | | | | | | | | | ✓ | | | ✓ | 2 (9.5) |
| BSc Pharmacy | | | | | | | | | | | | | | | | | | | ✓ | | | 1 (4.8) |
| Graduate (ie, non-medical degree) | | | | | | | | | | | | | | | | | | | | ✓ | | 1 (4.8) |
| Official position | | | | | | | | | | | | | | | | | | | | | | |
| Medical superintendent | | | | | | ✓ | | | | | | | | | | | | | | | | 1 (4.8) |
| Chief medical officer | | | | | | | | | ✓ | | | | | | | | | | | | | 1 (4.8) |
| Medical officer | | | | | | | | | | | | ✓ | | | | | | | | | | 1 (4.8) |
| Consultant | ✓ | | ✓ | | | | ✓ | ✓ | | | | | ✓ | ✓ | ✓ | ✓ | ✓ | | | | | 9 (42.9) |
| Surgeon | | ✓ | | | | | | | | ✓ | | | | | | | | | | | | 2 (9.5) |
| General medicine | | | | | | | | | | | ✓ | | | | | | | | | | | 1 (4.8) |
| Intern doctor | | | | ✓ | ✓ | | | | | | | | | | | | | | | | | 2 (9.5) |
| Ward nurse | | | | | | | | | | | | | | | | | | ✓ | | | ✓ | 2 (9.5) |
| Pharmacist | | | | | | | | | | | | | | | | | | | ✓ | | | 1 (4.8) |
| Medical records officer | | | | | | | | | | | | | | | | | | | | ✓ | | 1 (4.8) |
| Years of experience in position | | | | | | | | | | | | | | | | | | | | | | |
| <1 | | ✓ | | | ✓ | | | | | | | | | | | | | | | | | 2 (9.5) |
| 1–3 | | | | ✓ | | | | | | | | | | | | | | | | | | 1 (4.8) |
| 4–6 | | | | | | ✓ | | ✓ | | | | | | | | | | ✓ | | ✓ | ✓ | 5 (23.8) |
| 7–10 | | | ✓ | | | | ✓ | | | | | | | | | | | | ✓ | | | 3 (14.3) |
| >10 | ✓ | | | | | | | | ✓ | ✓ | ✓ | ✓ | ✓ | ✓ | ✓ | ✓ | ✓ | | | | | 10 (47.6) |
| Place of work | | | | | | | | | | | | | | | | | | | | | | |
| General hospital | | ✓ | ✓ | ✓ | | | | | | | | | | ✓ | ✓ | ✓ | ✓ | ✓ | ✓ | | | 9 (42.9) |

Continued

**Table 2** Continued

| Characteristics | 1 | 2 | 3 | 4 | 5 | 6 | 7 | 8 | 9 | 10 | 11 | 12 | 13 | 14 | 15 | 16 | 17 | 18 | 19 | 20 | 21 | n (%) |
|---|---|---|---|---|---|---|---|---|---|---|---|---|---|---|---|---|---|---|---|---|---|---|
| Regional hospital | ✓ | | | | | | | | ✓ | ✓ | ✓ | | ✓ | | | | | | | ✓ | ✓ | 7 (33.3) |
| Taluk hospital | | | | | | ✓ | | ✓ | | | | ✓ | | | ✓ | | | | | | | 5 (23.8) |
| Language(s) used during interview | | | | | | | | | | | | | | | | | | | | | | |
| English (only) | | ✓ | ✓ | ✓ | ✓ | ✓ | ✓ | ✓ | ✓ | ✓ | ✓ | ✓ | | ✓ | ✓ | ✓ | ✓ | | | | | 15 (71.4) |
| Hindi (only) | | | | | | | | | | | | | | | | | | | | ✓ | ✓ | 2 (9.5) |
| Malayalam (only) | | | | | | | | | | | | | | | | | | ✓ | ✓ | | | 2 (9.5) |
| English and Hindi (mixture) | ✓ | | | | | | | | | | | | ✓ | | | | | | | | | 2 (9.5) |

*Healthcare providers could select more than one answer to this question.

availability of healthcare staff compared with local healthcare facilities, such as smaller hospitals and primary/community health centres:

> We have very limited time, we did go to local hospital but doctors are not there. So if we get time we will come here rather than going to a hospital where there are no doctors. (IP 15)

However, multiple patients also reported that public hospitals were often crowded with high daily patient loads:

> There is so much crowd there you can't ask or hear anything there… so many people are there now, you cannot do anything. (IP 11)

The human resource issues at public primary and community healthcare facilities were also mentioned by HCPs:

> It will be useful if availability of doctors is ensured at the peripheral institutions around the clock. At times it is not there. (DOC 1)

Additionally, in our study settings most hospital doctors worked in both outpatient clinics and inpatient wards on a daily basis. Many doctors expressed concerns of time pressures due to the large patient volumes seen at hospital outpatient clinics and the subsequent lack of time they had to attend to all patients:

> We can hardly spend five minutes with each patient, seeing the crowd you will just want to finish everyone soon. (DOC 7)

Some doctors also reported that human and medical resource constraints across public healthcare facilities were hindering the quality of care that could be provided:

> [It's] not [about] motivation, [it's about] resource limitation. It's not humanly possible to see people every day for seven days. Quality definitely gets compromised. (DOC 3)

### Referral communication

A number of patients who recalled being referred from a previous healthcare facility to the hospital reported that they were not provided with any referral information:

> No, they didn't give any parchi [papers]. We were getting medicines right only that is with us. (IP 8)

HCPs also discussed referral communication. Doctors explained that there were no structured processes to follow for information exchange during referrals:

> Yeah there is no proper way of doing it… inpatients sometimes we have to [refer] but as I told you we never had a structured format. (DOC 14)

Despite the lack of structured systems, some doctors explained that they would make ad-hoc calls to ensure that some information was transferred when referring a patient.

**Table 3** Summary of charted data for inpatients (IPs)

| IP | Public healthcare service characteristics | | Healthcare provider to patient communication | | | Attitudes regarding medical information | | |
|---|---|---|---|---|---|---|---|---|
| | Large patient loads | Deficient primary care services | Verbal healthcare information during admission | Referral information | Impoliteness/ impatience | Transportation of medical documents | Patient-held booklet intervention | Dissatisfaction with lifestyle advice |
| 1 | | | | | | ✓ | | |
| 2 | | | ✓ | | | | | |
| 3 | ✓ | | ✓ | | ✓ | ✓ | | |
| 4 | | | ✓ | | | ✓ | | |
| 5 | | ✓ | ✓ | ✓ | ✓ | | | |
| 6 | | | | ✓ | | ✓ | | |
| 7 | ✓ | ✓ | | ✓ | | ✓ | | |
| 8 | | | | ✓ | | | | |
| 9 | | | ✓ | | | | | |
| 10 | | | ✓ | | | | | |
| 11 | ✓ | | ✓ | | | ✓ | | |
| 12 | | | ✓ | ✓ | | ✓ | | |
| 13 | | | ✓ | | | ✓ | | |
| 14 | | ✓ | ✓ | | | | ✓ | ✓ |
| 15 | | ✓ | ✓ | | ✓ | | ✓ | ✓ |
| 16 | | ✓ | ✓ | | | ✓ | ✓ | |
| 17 | | | ✓ | | | ✓ | ✓ | ✓ |
| 18 | | ✓ | ✓ | | | ✓ | ✓ | |
| 19 | | | ✓ | | | ✓ | ✓ | |
| 20 | | | | | | | ✓ | ✓ |

**Table 4** Summary of charted data for healthcare professionals (HCPs)

| HCP | System factors | | | Information exchange practices | | | | | Quality improvement strategies | | | |
|---|---|---|---|---|---|---|---|---|---|---|---|---|
| | Time and resource constraints | Absence of handover communication training | Absence of structured formats for information exchange between HCPs | Hospital record keeping | Ad-hoc phone calls | Patient-held medical documents | Discharge instructions | Hierarchical transfer of responsibility | Increase resource provision | Introduce formal referral systems | Implement 'e-health' systems | Patient-held booklet intervention |
| 1 | | | ✓ | | | ✓ | ✓ | | ✓ | | | |
| 2 | | | ✓ | | | ✓ | ✓ | | | | | |
| 3 | | | | | | ✓ | | | ✓ | ✓ | ✓ | |
| 4 | ✓ | | ✓ | | | ✓ | ✓ | | | | | |
| 5 | | | ✓ | ✓ | | ✓ | | | | | | |
| 6 | ✓ | | ✓ | | | | | | ✓ | | ✓ | |
| 7 | ✓ | | ✓ | | ✓ | | | | | | | |
| 8 | ✓ | | | | ✓ | | | | | | | |
| 9 | | | | | | | | | ✓ | | | |
| 10 | ✓ | ✓ | ✓ | | ✓ | | ✓ | | | | | |
| 11 | ✓ | ✓ | ✓ | | | | | | ✓ | ✓ | | |
| 12 | | | | ✓ | | ✓ | | | ✓ | | ✓ | |
| 13 | ✓ | | | | | | | | | | ✓ | ✓ |
| 14 | ✓ | | | | | | | | | | | ✓ |
| 15 | ✓ | | | | | ✓ | | | | | | ✓ |
| 16 | | | | | | | | | | | | ✓ |
| 17 | ✓ | | | | | ✓ | ✓ | ✓ | | | | ✓ |
| 18 | | | | ✓ | | | ✓ | ✓ | | | | |
| 19 | ✓ | | | ✓ | | | | | | | | |
| 20 | | | | ✓ | | | | | | | | |
| 21 | ✓ | | | | | | | ✓ | | | | |

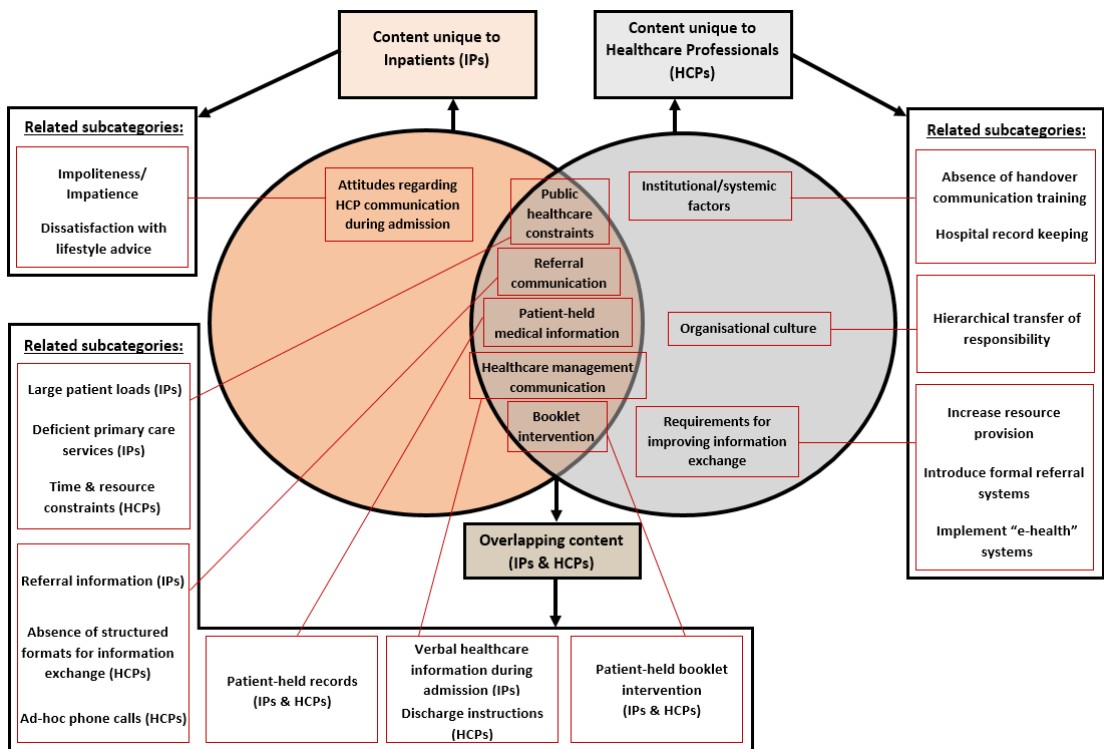

**Figure 1** Similarities and differences between the content of patient and healthcare provider data with related subcategories. IP, inpatients.

However, this appeared to depend on how well they knew the patient or doctor:

> Sometimes I call the doctor to tell them that so and so is coming. Please do the needful. If I know the patient or doctor. (DOC 11)

### Patient-held medical information

At the point of hospital admission, patient-held notes and/or medical records can facilitate optimal care by providing HCPs with key patient-specific information. When asked about whether they brought medical papers to the hospital, most patients reported that they regularly stored and transported papers to HCP visits; these included referral notes, prescription cards, test results and other records from inpatient/outpatient/primary care:

> Yeah we have always kept everything safely. [Shows researcher a bag with all sort of papers like reports, lab tests, etc.] (IP 3)

Doctors also talked about patient-held medical information during interviews. For example, some doctors reported that patients regularly kept and transported their medical records:

> Almost everyone comes with medical reports. (DOC 11)

However, other doctors described that, in their experience, the availability of patient-held records was less consistent and that this could have a negative impact on the continuity of care provided:

Some of them do bring investigations and all others don't bring much and we have to work out what happened from the start. (DOC 3)

### Healthcare management communication

When asked about verbal HCP communication, many patients reported that during admission and/or discharge, HCPs had provided them with some basic verbal healthcare management information (ie, medication, treatment, lifestyle and/or follow-up requirements). However, the quantity of information received appeared to vary notably between patients. For example, some recalled being given detailed instructions:

> Doctor says everything. I was given medicines and now they asked me to take injections also. Doctor is saying I am not controlling my sugar. The nurse taught me how to take injection. (IP 19)

Conversely, others appeared to receive relatively limited information, and one carer reported having to seek healthcare advice from alternative sources:

> Doctors don't explain everything. We speak to our friends and get details from them. (Carer-IP 16)

HCPs also discussed their healthcare communication practices with patients. While talking about discharge, a nurse explained that they predominantly provided verbal instructions and described the usual amount of time taken to explain information to each patient:

Usually we take twenty to twenty-five minutes to instruct the patients. If the patients understand then it can be even faster. (NUR 1)

Doctors reported that they provided patients with documented information on discharge cards and verbally advised patients to return to their local HCP/healthcare institution during the discharge consultation:

We give them a discharge card. Discharge card is there we have written and then we refer them to the local hospital or where they come from. (DOC 15)

### Booklet intervention

During interviews that took place after the handover expert meeting, patients were asked for their opinion regarding the utility of a patient-held booklet where medical documents could be stored, organised and transported to HCP visits. Most appeared to think that it could be effective and help with self-management, including those who were illiterate:

Yeah, sometimes we don't know what to do so it would be good if some paper is there to help us. We can't read it ourselves but our son or daughter-in-law can help us. (IP 17)

HCPs were also asked for their opinions regarding the booklet intervention. Many generally felt it could be useful, but various conditions and/or reservations were also expressed. For example, doctors felt that the success of the booklet would rely on patient attitudes:

That will depend on the patients, if they maintain that and bring it every time. For us there is no change, we write our observations in paper or notebook, doesn't matter... Might be helpful. (DOC 22)

Related to this, one doctor felt that in order to see the most benefit, patients needed to be regularly instructed to keep and transport their medical documents:

We write the communication but the patients don't keep them proper. I think we have to tell the patients to keep the letters and papers. (DOC 4)

### Content unique to patients

#### Attitudes regarding HCP communication during admission

A few patients recalled receiving some impolite and/or impatient treatment from healthcare staff during their hospital admission:

The doctors don't speak much. They explain but get angry if you don't understand them. (IP 3)

In addition, some patients expressed dissatisfaction with the lifestyle advice provided. In particular, patients of lower socioeconomic status felt that nutritional instructions were not suitable for them due to time and financial constraints:

We are daily labourers we can't follow all the instructions... We can't follow that, we are poor we do

hard work and we just can't concentrate on eating. Whatever is there we just eat. (IP 15)

### Content unique to HCPs

#### Institutional/systemic factors

Some doctors displayed good knowledge of the key information that should be transferred during patient referrals, transfers and/or hospital discharge:

To another hospital, yeah first we have to write what are the main complaints of patients presenting illness and write about the past history, then we will write about what all investigations we have done here 'til the day of transfer, then what is the condition of the patient we are discharging, why we are discharging (and) any investigations, major investigations, to be done. (DOC 2)

However, when asked about training opportunities, numerous doctors mentioned that they had not received any formal handover training. Some recalled that this type of training was not provided at medical school:

I think it was not there in medical curriculum. (DOC 1)

Others reported that training was not provided in their workplace/s and instead they learnt on the job:

We are sent to the wards, we see what our seniors do and we do that's all. We have to develop our communication skills ourselves no formal training is there. (DOC 14)

When asked about hospital record keeping, a medical records officer stated that inpatient records are stored in hospitals for up to 10 years following patient discharge. However, the same officer also indicated that these paper-based records are not easily accessible:

Definitely I can locate any record but it might take some time to locate them. (MRO 1)

#### Organisational culture

Based on reports from both doctors and nurses, it appeared as though some hierarchical transfer of responsibility for documented handover and healthcare communication took place in hospitals. For example, a senior doctor mentioned that they instructed medical interns to write notes for them when their patient load was high:

We do write in the papers, whether it's discharge card or outpatient sheets. When patient load is high, then we tell our interns to do it for us, we check that and then sign. (DOC 22)

#### Requirements for improving information exchange

During interviews, HCPs were asked for their thoughts on requirements to improve information exchange between HCPs and between HCPs and patients. Numerous doctors

felt that there needed to be a notable increase in public healthcare resource provision:

> Infrastructure is very small but the outpatient department is ten times more than it can manage, so more posts should be created… We have to increase the manpower and also our materials. (DOC 15)

Some doctors also discussed the idea of introducing standardised referral documents and systems to improve referral communication:

> You can supply people with [referral] forms and make it mandatory that residents have to maintain a register. In that case they will maintain the register. (DOC 3)

In addition, while discussing current information systems, one doctor in Kerala reported that an application had been made for a near-future transition to computerised healthcare information systems. This appeared to be a state-wide plan for public healthcare facilities:

> We have submitted a proposal for paperless computerisation system for doctors, so I think state-wide they are planning to do that. (DOC 6)

## DISCUSSION
### Main findings

This study presents qualitative data on patient and HCP knowledge, attitudes and barriers to handover and healthcare communication during public hospital inpatient care in the states of Kerala and Himachal Pradesh in India. The main finding is that verbal and documented information exchange between HCPs and between HCPs and patients is often suboptimal during referrals, hospital admission and discharge, with a lack of structured systems and HCP education in place to ensure sufficient continuity of care. While unique themes emerged for both patients and HCPs, a comparison of the results from each participant group showed that there was also a notable amount of overlapping content. The results have highlighted the challenging and multifaceted nature of handover and healthcare communication during inpatient care in India. With regard to public health, the findings have also elucidated a number of key areas to address to improve the continuity and safety of chronic NCD patient care.

Some of the results from the current study reflect and reinforce findings from previous research focusing on outpatient care in the same study areas of India.[20] In particular, during interviews in both studies, patients and HCPs recognised the resource constraints affecting public healthcare. The main issue reported was deficient primary healthcare services, which is in line with well-established findings of limited primary care infrastructure across India and numerous LMICs.[36] In our study settings, under-resourced primary care resulted in many patients preferring to visit hospitals as the first point of

care. Subsequently, large patient loads were seen in both outpatient and inpatient departments, which limited HCP consultation times. Other key areas of discussion in the current study reflected in the outpatient findings were inconsistent transportation of patient-held medical documents and views regarding the utility of patient-held booklets. While more inpatients than outpatients reported that they regularly transported records to HCP visits, some doctors recalled seeing many patients who did not bring information to the hospital. This was problematic because if patients did not bring their records then doctors had to gather details from scratch, potentially compromising their continuity of care. When asked about the possible utility of introducing patient-held booklets to store and transport medical documents, inpatients had similar views to outpatients, which were generally positive, but also felt that the inclusion of self-management information would be beneficial. Doctors in the current study expressed a wider variety of views regarding booklets, but broadly thought that they could be useful if patients had positive attitudes towards their maintenance and use.

Regarding referral communication, the current study also highlighted similar issues of deficient information exchange between levels of care observed in the previous outpatient study.[20] For example, reports from both HCPs and patients revealed that that documented information was often provided in the form of minimal, handwritten notes on papers provided for other purposes (eg, prescription cards). These findings reflect results from other LMIC studies that have evidenced the exchange of poor-quality referral documents.[14 37–39] However, the current study also evidenced patient reports of not being provided with any documented information during referrals. Further, while a small number of inpatient doctors in the current study explained that they called HCPs to discuss a referral case, this appeared to be dependent on how well they knew the patient or HCP. Such findings indicate that there are further inconsistencies in referral communication practices than previously described. Overall, these deficits are unsurprising given that multiple HCPs in both the current and previous outpatient study reported an absence of structured systems and education provided for handover communication. These findings are also in line with the few previous descriptions from India of a paucity of training and protocols for handover practices.[15–17]

In addition to similarities found with previous research, the current study has elucidated numerous novel insights regarding handover and healthcare communication during critical points in inpatient care, which were previously unexplored in the study areas of India. Regarding inpatient medical record keeping, a records officer indicated that hospital records were not easily accessible when reporting that retrieving a specific record from storage could take "some time". Alongside the inconsistent transportation of patient-held records, this limited accessibility of medical information carries notable risks for patient safety. This is because,

without timely key patient background and/or treatment details, critical oversights can be made that result in adverse events.[4][5][7] Additionally, there were notable variations in patient reports of the provision of healthcare management information during hospital admission and discharge; while some patients reported being given clear self-care instructions, others stated that they sought information from external sources due to the lack of detail provided by hospital HCPs. It appears that the time pressures experienced by HCPs were a significant contributory factor to inconsistencies in HCP to patient communication, particularly at the point of discharge. During interviews, multiple HCPs reported often being busy with high patient loads and it was explained that the duty of writing discharge notes was passed from senior doctors to interns or nurses during busy periods. Additionally, it seemed that more time was spent on verbal discharge communication, with a nurse reporting that they typically took around twenty minutes per patient to explain discharge instructions. Such practices may be compromising the retention of key healthcare information, as global literature suggests that patients can struggle to absorb verbal details provided during consultations.[40] The potential implications of these findings are significant, given the associations that have been found between deficient discharge communication and an increased likelihood of adverse events.[7–10]

Furthemore, a key issue affecting handover and healthcare communication mentioned solely by patients was the receipt of impolite and/or impatient treatment from hospital doctors during admission. A small number of patients were also dissatisfied with the take-home nutritional advice provided, as they felt it failed to take into account their socioeconomic deprivation. These results may be explained by the reported lack of communication training in medical education, as well as a historical tendency for paternalistic physician conduct in India.[41] In other areas of India and Asia, research on HCP–patient communication has evidenced asymmetric power balances and patient dissatisfaction during patient consultations.[42] Such findings reveal the need for more patient-centred communication, particularly for poorer patients, who make up a significant proportion of public healthcare users in India. As for requirements for improvement reported by HCPs, during interviews many doctors recognised the need for an increase in public healthcare resource provision, as well as structured systems for information exchange. Some also discussed the promise of implementing 'e-health' systems, with a doctor in Kerala reporting that public healthcare facilities across the state will be transitioning to computerised systems. While our colleagues from Kerala report that this development is in its early stages, it holds potential as similar systems in HICs and other LMICs have helped to advance information accessibility and the overall quality of healthcare provided.[43][44]

## Strengths and limitations

As far as the authors are aware, this is the first study to qualitatively explore both patient and HCP knowledge, attitudes and barriers to multiple areas of handover and healthcare communication during chronic NCD inpatient care in India. The use of qualitative methodology and inclusion of multiple healthcare sites have revealed a number of key issues that are reflected among the emerging LMIC literature, suggesting likely transferability to other settings. Interviews with both patients and HCPs have provided a variety of valuable perspectives, which has helped to identify critical areas impacting the continuity of chronic NCD inpatient care. The number of interviews conducted helped to achieve data saturation for both participant groups and study credibility was strengthened via the use of multianalyst triangulation.[34]

The lack of documented inclusion/exclusion rates for participation is a limitation, as this could not be recorded. In addition, the accuracy of recall of the minority of patients interviewed at home may have been limited by the delay between recruitment and data collection. Recruitment challenges meant that patient participants were predominantly older (ie, 45+ years), which limited exploration of younger patient experiences; this was, however, largely unsurprising given that the study exclusively recruited patients with chronic NCDs. The cross-cultural nature of this research may have resulted in constraints during data collection and analysis, as ingroup bias could have affected participants' willingness to openly converse with a non-local researcher.[45] Social desirability bias from the use of individual interviews and participants' awareness that the interviewer was a public health professional may have also affected truthfulness of the data.[46] Despite these challenges, the recurrence of themes indicating data saturation and the finding that our results are supported by existing literature suggest that they had minimal impact.

## Conclusions and next steps

This study has found that handover and healthcare communication for inpatients with chronic NCD during referrals, hospital admission and discharge is often fragmented. The critical barriers appear to be a lack of structured information exchange systems and HCP education. There is also a growing recognition of the need for the government to strengthen primary healthcare infrastructure in line with the Declaration of Alma-Ata.[47] This will greatly assist in increasing accessibility of care and subsequently reducing pressure on hospital services. It will also be required to address the United Nations' sustainable development goals regarding universal health coverage and reducing premature deaths from NCDs.[48] In addition, the implementation of structured documentation, systems and training is urgently required to manage critical care transitions such as referrals, transfers and discharge. Research from both HIC and LMIC settings has proven that such interventions can improve the continuity and safety of care.[4][17][37][49] Regarding future steps, during HCP

interviews it was reported that public healthcare facilities in Kerala will be transitioning to computerised 'e-health' information systems. The Indian government has also since pledged to digitise all public healthcare information systems in the country via an 'Integrated Health Information Platform'.[50] While such developments hold promise and are progressing, they remain in their initial stages in many states and face numerous infrastructural challenges. Additionally, they are not likely to target issues regarding HCP to patient communication, patient access to healthcare information and information exchange between public and private HCPs.

Therefore, a mixed-methods pilot study exploring the design and implementation of patient-held record booklets is suggested. This could ameliorate some of the current issues by incorporating disease-specific and structured documents, which have been shown to improve the recording of clinical information and can provide a means of organising records in a logical and accessible way.[49 51 52] The patient-held nature of this strategy could also increase patient access to key healthcare information, which may improve self-management. Given the unstructured, predominantly paper-based systems used across the study sites, this is an area for development that has been welcomed by Indian national and international experts, as well as by patients and HCPs in our study areas. There have also been multiple international successes of improved continuity of care via utilisation of similar patient-held/home-based records in both outpatient and maternal and child healthcare.[26–30 53] In order to maximise booklet utilisation, it would be necessary to address the issues surrounding patient retention and understanding of the importance of medical documents. Initial key steps could be to involve both patients and HCPs in the design process and accompany the introduction of booklets with relevant promotion, training and incentives.

Finally, given the rising burden of NCDs across LMICs, this research is timely and crucial for effective health systems development. Further LMIC research is required to continue exploring the critical factors affecting handover, continuity of care and health systems integration and to develop sustainable and cost-effective interventions.

**Author affiliations**
[1]Institute of Applied Health Research, University of Birmingham, Birmingham, UK
[2]Centre for Chronic Disease Control, Gurgaon, Haryana, India
[3]Public Health Foundation of India, New Delhi, India
[4]Sree Chitra Tirunal Institute of Medical Sciences and Technology, Trivandrum, Kerala, India
[5]Amrita Institute of Medical Sciences, Kochi, Kerala, India
[6]Academic Unit of Primary Care, University of Warwick, Coventry, UK
[7]Centre for Applied Health Research and Delivery, University of Warwick, Coventry, UK

**Acknowledgements** This research was supported by the National Institute for Health Research (NIHR) Collaboration for Leadership in Applied Health Research and Care West Midlands (CLAHRC WM). We would like to extend our thanks to all patients, healthcare staff and researchers who kindly took the time to participate in this project. We are also indebted and give thanks to the participating healthcare facilities. Finally, we are very grateful to the Directors of Health from both Himachal Pradesh and Kerala states in India for their assistance in facilitating this project. This research would not have been possible without their support.

**Contributors** CH: data curation, formal analysis, visualisation, writing (both original draft and final review and editing). SJ: investigation, data curation, project administration, writing (review and editing). JP: conceptualisation, funding acquisition, project administration, supervision, writing (review and editing). SKS: funding acquisition, project administration, supervision, writing (review and editing). SGo: funding acquisition, projection administration, supervision, writing (review and editing). PD: conceptualisation, funding acquisition, project administration, supervision, writing (review and editing). PG: conceptualisation, data curation, funding acquisition, writing (review and editing). SGr: funding acquisition, formal analysis, writing (review and editing). RL: conceptualisation, funding acquisition, writing (review and editing). SM-H: conceptualisation, funding acquisition, investigation, methodology, project administration, supervision, writing (review and editing).

**Funding** This research was jointly funded by the Department for International Development, the Economic and Social Research Council, the Medical Research Council and the Wellcome Trust (grant number: MR/M00287X/1).

**Competing interests** None declared.

**Patient consent for publication** Obtained.

**Ethics approval** This study was reviewed and approved by the Centre for Chronic Disease Control Independent Ethics Committee, India, and the Amrita Institute of Medical Sciences Institutional Ethics Committee, India. Data archives will be stored at the University of Birmingham, in accordance with the University's code of practice.

**Provenance and peer review** Not commissioned; externally peer reviewed.

**Data availability statement** Data are available upon reasonable request.

**ORCID iD**
Claire Humphries http://orcid.org/0000-0001-9784-5218

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
