## [Reviewer comments · BMJ Open]

ARTICLE DETAILS

TITLE (PROVISIONAL)	Patient and healthcare provider knowledge, attitudes and barriers to handover and healthcare communication during chronic disease inpatient care in India: A qualitative exploratory study
AUTHORS	Humphries, Claire; Jaganathan, Suganthi; Panniyammakal, Jeemon; Singh, Sanjeev; Goenka, Shifalika; Dorairaj, Prabhakaran; Gill, Paramjit; Greenfield, Sheila; Lilford, Richard; Manaseki-Holland, Semira

VERSION 1 – REVIEW

REVIEWER	Hanneke Merten, PhD, lecturer and researcher quality and safety in healthcare Amsterdam UMC, location VUmc, department of Public and Occupational Health The Netherlands
REVIEW RETURNED	23-Dec-2018

GENERAL COMMENTS	Major points • The authors have a recently published paper which reports on a quite similar topic (Humphries et al. Investigating clinical handover and healthcare communication for outpatients with chronic disease in India: A mixed-methods study. PLoSOne. 2018;13(12):e02070511). Throughout the manuscript this paper is mentioned and referenced, but being familiar with the content of that paper, I find it difficult to decide on the added value of the current paper. If I understand it correctly, the recently published paper focusses on outpatients while the current paper focusses on inpatients. However, there appear to be many similarities, such as the methodological approach for the interviews, the meeting halfway through the study with the same participants, the choice for the patient held-booklet and the themes derived from the data. This is why I also struggle to answer question 14 in the review, it is just not clear enough for me what this papers adds in addition to the recently published manuscript, but that may also just be based on my misunderstanding.• Abstract, conclusion, line 52-53. The authors mention that “a well-designed patient-held record booklets may be an acceptable and effective part of the solution’. However, in the results section of the abstract I cannot see any findings supporting this statement. The only mentioning in the abstract is in lines 45-47, but that is on patient-held documents in general. This point also comes back in the main manuscript. Although the choice for the patient-held booklet appears to be based on the meeting halfway through the study, I find it difficult to see how this recommendation follows from the findings and why this would be the most appropriate intervention.
---

	 • Introduction/methods: did the authors use any type of theoretical model for the development of the topic guides and interpretation of the data? • Methods, p7, 2.3 lines 157-160: Busy healthcare providers were not included for the interviews. How often did the research-assistants approach these HCPs? And if they are not included, how can the authors be confident they have reached data saturation? The viewpoint of these extremely busy HCPs is missing, this could have led to a selective sample. Do you know how many were unavailable? Also, in the discussion (29, lines 33 and further) it is mentioned that predominantly older patients were included, leading to missing viewpoints of younger patients. How could data saturation have been reached? • Methods, p8, lines 168-169: The interviews took place either at the hospital ward or at home five weeks after discharge. This raises two questions with me: first, how was patient confidentiality ensured when interviewing on a busy hospital ward? Second, why was there such a long time period when the patient could not be interviewed at the ward, five weeks after discharge seems like a long time, how are they comparable? Or were there two separate interviews? • Methods, p9, line 209: The interviews were conducted in India, translated and analysed by a researcher not involved in the data collection and based in Great-Britain. Would the translation have influenced the findings? For the interpretation of qualitative data, familiarity with the local context is important. How was the researcher able to interpret the data for the local context? I have doubts whether consulting with the local researcher would be enough (how often) to familiarise with the local context. Could this have influenced the findings? • Methods, p10, line 220: The qualitative data were analysed without the use of qualitative software, only Microsoft Excel was used to categorize and organise the themes. I wonder why the authors have made this choice and if this has potential consequences for the findings? Especially in the stages of coding, reorganising, categorising, charting and interpreting the findings this may hold the risk of missing potentially important themes and statements. • Results, p23, line 46 and further: "Despite displaying good knowledge of what information should be transferred...": how did the authors establish this, were the doctors observed during patient handovers? And what standard was used? • Results, p25, lines 14-19: Although I understand the need for additional manpower, I have doubts whether this should be labelled as an intervention to improve intervention exchange. When reading the quote, this feels more as a prerequisite for any intervention to be potentially successful. • Results, p25, lines 32-43: The statement that a computerised system for information exchange was developed raises questions about the potential added value of the patient-held booklet, especially since it was mentioned that a substantial part of the patients involved did not bring their patient-held documentation to the hospital at all. • Discussion, p27, lines 47 and further: Why do the authors expect that using a patient-held booklet would work better than the patient-held documents so far? Minor points
--	--

	 • Abstract, p2, line 34: abbreviation NCD should be explained first time and abbreviation HPC should be HCP • In the strengths and limitations it is mentioned that “this is the first qualitative study...”. But a recently published paper of the authors (Humphries et al. Investigating clinical handover and healthcare communication for outpatients with chronic disease in India: A mixed-methods study. PLoSOne. 2018;13(12):e02070511) reports on a comparable topic, although with a focus on outpatients. • Introduction, p5, line 107: please use consistent format for referencing • Results: Table 2: Explain abbreviations MPH and MBBS. • Data sharing statement: for qualitative studies, the transcripts can be made available, please state whether this is the case or not (and why).
--	--

VERSION 1 – AUTHOR RESPONSE

Reviewer 1 – Major comments:

1. The authors have a recently published paper which reports on a quite similar topic (Humphries et al. Investigating clinical handover and healthcare communication for outpatients with chronic disease in India: A mixed-methods study. PLoSOne. 2018;13(12):e02070511). Throughout the manuscript this paper is mentioned and referenced, but being familiar with the content of that paper, I find it difficult to decide on the added value of the current paper. If I understand it correctly, the recently published paper focusses on outpatients while the current paper focusses on inpatients. However, there appear to be many similarities, such as the methodological approach for the interviews, the meeting halfway through the study with the same participants, the choice for the patient held-booklet and the themes derived from the data. This is why I also struggle to answer question 14 in the review, it is just not clear enough for me what this papers adds in addition to the recently published manuscript, but that may also just be based on my misunderstanding.

Thank you for your comments. We have responded to the points made in the sections below:

How the study relates to the published paper:

- This study is separate and forms part of a project that is made up of multiple studies (including the outpatient study) covering different aspects of handover and healthcare communication within Himachal Pradesh and Kerala states, India. Therefore, whilst data from inpatients was collected within the same time period as that collected from outpatients, it was collected independently (i.e. within different care settings and from different participants) and using different interview topic guides.
- With regard to the healthcare provider (HCP) data, instead of conducting separate interviews with HCPs regarding inpatient and outpatient care, interviews covered aspects of handover and healthcare communication affecting each type of care. This is because the majority of HCPs in the study settings worked with both outpatients and inpatients on a daily basis. Therefore, interviews (n=17) from the previous outpatient study involving HCPs that worked with both inpatients and outpatients were also analysed for the inpatient study (i.e. this study involved secondary analysis of HCP data). Also, a small number (n=4) of additional HCP interviews (also collected within the same period and using the same topic guide) were analysed for the inpatient study only, as they came from HCPs solely involved in inpatient care.
- For this inpatient study, all data was analysed using an inductive framework method that focussed on extracting categories and themes relevant to handover and healthcare communication during inpatient care.

What this paper adds to the recently published outpatient manuscript:

- The similarities in results regarding systemic and patient-related issues affecting handover and healthcare communication provide stronger evidence that these challenges are indeed commonplace and affect both outpatient and inpatient care within the study settings.
- The current study focussed purely on handover and healthcare communication during inpatient care, which has not yet been investigated within the study settings. Inpatient care is particularly critical, as previous literature (predominantly from high-income nations) has repeatedly indicated that poor communication during inpatient care transitions and hospital discharge can have significant consequences for the continuity and safety of patient care.
- The current study has elucidated both patient and HCP-related factors affecting quality of handover and healthcare communication during inpatient care and particularly during hospital discharge. The solely qualitative nature of the study has enabled a more detailed exploration of these factors and has highlighted critical issues, such as factors related to HCP organisational culture and HCP to patient communication styles, which were not explored in the outpatient study.
- As well as describing further HCP and patient views regarding the idea of a patient-held booklet, the current study has also described novel HCP views regarding requirements for improving information exchange in the study areas. Therefore, it has strengthened the evidence regarding ideas and attitudes of context-specific solutions to issues of handover of healthcare communication.

Some of the above information has now been clarified in the introduction (see page 5, from line 116) and methods (see page 9, from line 204) sections of the manuscript. In addition, the discussion section (see page 29, from line 522) of the manuscript has been restructured to more clearly explain a) the similarities in findings with the previous outpatient manuscript and b) the novel discoveries that have been made and the added value they provide to the literature base.

2. Abstract, conclusion, line 52-53. The authors mention that “a well-designed patient-held record booklets may be an acceptable and effective part of the solution”. However, in the results section of the abstract I cannot see any findings supporting this statement. The only mentioning in the abstract is in lines 45-47, but that is on patient-held documents in general. This point also comes back in the main manuscript. Although the choice for the patient-held booklet appears to be based on the meeting halfway through the study, I find it difficult to see how this recommendation follows from the findings and why this would be the most appropriate intervention.

Thank you for your comments. The abstract has now been amended (see page 2, from line 39) to more accurately reflect the key findings of the study and the conclusions drawn. Further details regarding the choice for patient-held booklets as a suggested intervention have been provided in response to the comments numbered 12 & 13 below. Additional clarification has also been provided in the methods (see page 11, from line 255) and discussion sections (see page 30, from line 547) of the manuscript.

3. Introduction/methods: did the authors use any type of theoretical model for the development of the topic guides and interpretation of the data?

Thank you for your question. We have responded in the sections below:

Topic guide development:

- Topic guides were developed using relevant handover literature and local knowledge of health systems functioning within the study areas. They were then piloted over three rounds prior to commencement of data collection to ensure they were clear as well as culturally and contextually appropriate.

Interpretation of the data:

- Due to the exploratory and descriptive nature of the study, data was analysed using an inductive thematic analysis approach utilised in Grounded Theory, which focuses on the identification of concepts that emerge from study interviews, as opposed to being guided by a pre-existing theoretical framework.
- The choice of approach for the interpretation of data was guided by previous qualitative studies within the same topic area (i.e. handover and continuity of care) from both high and low-income countries that have used similar methods. A few examples of such studies include:
 - o Hesselink G, Flink M, Olsson M, Barach P, Dudzik-Urbaniak E, Orrego C, Toccafondi G, Kalkman C, Johnson JK, Schoonhoven L, Vernooij-Dassen M. Are patients discharged with care? A qualitative study of perceptions and experiences of patients, family members and care providers. *BMJ Qual Saf.* 2012 Dec 1;21(Suppl 1):i39-49.
 - o Flink M, Öhlén G, Hansagi H, Barach P, Olsson M. Beliefs and experiences can influence patient participation in handover between primary and secondary care—a qualitative study of patient perspectives. *BMJ Qual Saf.* 2012 Dec 1;21(Suppl 1):i76-83.
 - o Sarvestani R, Moattari M, Nasrabadi AN, Momennasab M, Yektatalab S. Challenges of nursing handover: a qualitative study. *Clinical nursing research.* 2015 Jun;24(3):234-52.

Some of the above information has now been included in the methods section (topic guides - see page 10, from line 234 and interpretation - see page 12, from line 282) of the manuscript for further clarification.

4. Methods, p7, 2.3 lines 157-160: Busy healthcare providers were not included for the interviews. How often did the research-assistants approach these HCPs? And if they are not included, how can the authors be confident they have reached data saturation? The viewpoint of these extremely busy HCPs is missing, this could have led to a selective sample. Do you know how many were unavailable?

Thank you for your questions. Research assistants approached HCPs on a daily basis in order to recruit study participants. If HCPs stated that they were too busy on a given day, a note was kept and they were not approached again. However, this did not exclude them from the study entirely, so if they were approached and available on another day then they were able to participate. This has now been clarified in the methods section of the manuscript (see page 8, from line 193).

Due to the busy and dynamic nature of opportunistic data collection for this study, a record could not be kept by researchers of how many HCPs did not participate at all due to being too busy/unavailable. However, the lead researcher (SJ) informed us that this number was small, as HCPs were generally intrigued by the research and subsequently motivated to participate.

Overall, given the number of HCPs with various experience and expertise recruited and the similarities in codes and themes developed from their transcripts, the authors remain confident that data saturation was reached. However, we have now acknowledged a lack of adequately recorded inclusion/exclusion rates as a limitation of the study (see page 33, from line 631).

5. Also, in the discussion (29, lines 33 and further) it is mentioned that predominantly older patients were included, leading to missing viewpoints of younger patients. How could data saturation have been reached?

Thank you for your question. The lack of data from younger participants could indeed be a potential limitation. However, this limitation appears overstated in the absence of an explanation that it is unsurprising that most patients were older, given the study exclusively recruited those with chronic, non-communicable diseases. Based on this information, as well as the number of patients recruited and the similarities in codes and themes developed from transcripts, the authors remain confident that data saturation was reached. Some of the aforementioned information has now been clarified in the limitations section (see page 33, from line 634) of the manuscript.

6. Methods, p8, lines 168-169: The interviews took place either at the hospital ward or at home five weeks after discharge. This raises two questions with me: first, how was patient confidentiality ensured when interviewing on a busy hospital ward?

Thank you for your question. Given the lack of available private spaces to interview patients within public healthcare facilities, there was no other option but to interview patients on hospital wards. In general, inpatient wards were not largely overcrowded and so interviews were able to be completed in a relatively quiet and private manner. All participants were comfortable with the nature of the interviews and provided full consent to participate prior to data collection. In addition, the questions asked were not sensitive in nature and HCPs were not present during inpatient interviews. Some additional information has been provided in the methods section (see page 9, from line 221) for further clarity.

7. Second, why was there such a long time period when the patient could not be interviewed at the ward, five weeks after discharge seems like a long time, how are they comparable? Or were there two separate interviews?

Thank you for your question. A small number of patients (n=4) were interviewed either at 5 weeks (n=2) or 4 months (n=2) following discharge for their convenience, as during recruitment they were in the process of being discharged and leaving hospital. The specific follow-up times coincided with community visits being completed for another quantitative study within the India handover project, which these patients also participated in. This information has now been clarified in the methods section (see page 9, from line 224) of the manuscript.

All interviews are comparable as they each relied on patient recall and used the same topic guide. The only difference in questioning for home-based interviews was the way in which healthcare utilisation questions were framed - i.e. home-based interviews asked about post-hospital healthcare utilisation rather than pre-hospital utilisation - which did not affect the objectives of the current study. However, it is recognised that the lag in time for the small number of follow-up interviews may have reduced the reliability of patient recall. Therefore, this has now been noted in the limitations section (see page 33, from line 632) of the manuscript).

8. Methods, p9, line 209: The interviews were conducted in India, translated and analysed by a researcher not involved in the data collection and based in Great-Britain. Would the translation have influenced the findings? For the interpretation of qualitative data, familiarity with the local context is important. How was the researcher able to interpret the data for the local context? I have doubts whether consulting with the local researcher would be enough (how often) to familiarise with the local context. Could this have influenced the findings?

Thank you for your questions. We have responded in the sections below:

Translation:

- We would like to note that the majority of both inpatient and HCP interviews were conducted purely in English and so did not require translation (see bottom sections of table 1, page 15 and table 2, page 17)
- For the remaining minority of interviews, all translations were completed by a researcher familiar with the local context and fluent in all the local dialects as well as English.
- In addition, before being sent to the UK researcher, a qualitative expert and co-author based in India crosschecked all translations. This expert was also familiar with the local context and fluent in all the languages utilised during interviews.
- Given the above information, the authors remain confident that the translation involved in this study has not had a detrimental impact on the findings.

Familiarity with local context and interpretation:

- Whilst the lead UK researcher (CH) was not present during data collection, they became familiar with the local context by visiting the study sites in both Himachal Pradesh and Kerala states, India. The Himachal Pradesh site visits took place shortly after participation in the handover expert meeting in New Delhi and the Kerala site visits took place during an internship with the State Department of Health and Family Welfare.
- It was in addition to site visits that CH consulted with the lead Indian researcher (SJ) during the data familiarisation period to avoid any misinterpretations of the data.
- In addition, CH consulted with another qualitative expert co-author based in the UK, who has a wealth of experience in cross-cultural research and therefore was able to provide suitable advice.

Additional information regarding some of the above information has now been included in the methods section (see page 9, lines 214 – 220 and page 12, from line 273) of the manuscript.

9. Methods, p10, line 220: The qualitative data were analysed without the use of qualitative software, only Microsoft Excel was used to categorize and organise the themes. I wonder why the authors have made this choice and if this has potential consequences for the findings? Especially in the stages of coding, reorganising, categorising, charting and interpreting the findings this may hold the risk of missing potentially important themes and statements.

Thank you for your comments. Whilst the lead researcher (CH) was familiar with the essentials of qualitative software at the time of analysis, the decision was made to use Microsoft Excel to organise the data as this is what has been previously and successfully used for a similar method of inductive qualitative analysis in the outpatient study. Ultimately, any software used is simply a system for storing/managing the data that bears no consequence for analysis, which is an intellectual exercise. This point is demonstrated in the following quote:

“The key message is that unlike statistical software, the main function of CAQDAS is not to analyse data but rather to aid the analysis process, which the researcher must always remain in control of. In other words, researchers must equally know that no software can analyse qualitative data. CAQDAS are basically data management packages, which support the researcher during analysis”

Reference: Basit T. Manual or electronic? The role of coding in qualitative data analysis. Educational research. 2003 Jun 1;45(2):143-54.

Therefore, utilisation of Microsoft Excel did not eliminate the need to think and deliberate, generate codes and reject and replace them with others that were more illuminating. In addition, analyst triangulation took place with a qualitative expert and co-author during coding to ensure reliability in the analytical process.

10. Results, p23, line 46 and further: “Despite displaying good knowledge of what information should be transferred...”: how did the authors establish this, were the doctors observed during patient handovers? And what standard was used?

Thank you for your questions. There are multiple quotes from doctors listing out the key information that should be transferred during patient handovers at the point of transfers/referrals and/or discharge - this information was not included within the original manuscript in an attempt to keep the results section as concise as possible whilst addressing the principal findings. The standard used to interpret the level of HCP knowledge was based upon relevant literature that describes what information should be transferred during patient care transitions. Example of relevant literature include:

- o Newton, J., Eccles, M. and Hutchinson, A., 1992. Communication between general practitioners and consultants: what should their letters contain?. *Bmj*, 304(6830), pp.821-824.
- o Beaglehole R ,. Improving the prevention and management of chronic disease in low-income and middle-income countries: a priority for primary health care. *Lancet* 2008;372:940–9
- o Van Walraven C, Rokosh E. What is Necessary for High-Quality Discharge Summaries?. *Am J Med Qual*. 1999;14(4): 160–9. pmid:10452133
- o Coleman EA. Falling through the cracks: challenges and opportunities for improving transitional care for persons with continuous complex care needs. *J Am Geriatr Soc*. 2003;51(4): 549–55. pmid:12657078

To improve clarity, the results section (see page 26, from line 470) now includes a relevant verbatim quote from a doctor.

11. Results, p25, lines 14-19: Although I understand the need for additional manpower, I have doubts whether this should be labelled as an intervention to improve intervention exchange. When reading the quote, this feels more as a prerequisite for any intervention to be potentially successful.

Thank you for your comments. The first sentence within this section should in fact say, “HCPs were asked for their thoughts on requirements for improving information exchange” (as is mentioned in the heading above it), rather than specifically referring to intervention ideas only. This has now been corrected in the results section (see page 28, from line 504) of the manuscript.

1. Results, p25, lines 32-43: The statement that a computerised system for information exchange was developed raises questions about the potential added value of the patient-held booklet, especially since it was mentioned that a substantial part of the patients involved did not bring their patient-held documentation to the hospital at all.

Thank you for your comments. We have responded to the points made in the sections below:

Added value of patient-held booklets:

- Whilst computerised systems offer the promise of integrated and efficient systems for information transfer, these developments remain in their early stages and face a number of technical challenges prior to becoming fully functional.
- In addition, whilst e-health implementation is being rolled out relatively quickly and on a wide scale in Kerala, the situation is not the same in Himachal Pradesh and most other states of India. Therefore, there is still much need for other cost-effective and accessible methods of ensuring continuity of care – particularly for patients with chronic NCDs.
- Further and perhaps most importantly, the implementation of e-health systems will not necessarily address improving information exchange between HCPs and patients. Therefore, patient-centred strategies such as patient-held booklets warrant further investigation.

- The patient-held booklet intervention was suggested based on findings from relevant international research literature of successful use of home-based records for outpatients and in child and maternal healthcare. Also, during the course of our study experts from numerous international and Indian health systems experts recognised the potential benefit of this tool alongside the gradual development of computerised systems.
- One reason for this is because within our study areas many patients choose to visit both public and private healthcare providers, who utilise different information systems. Therefore, patient-held booklets were suggested as tools that could also help to improve the flow of critical information between such providers. Another reason was that booklets could contain structured, clear and comprehensible information for patients, which could lead to improved self-management of chronic conditions.

Addressing lack of patient transportation of medical documents:

- Findings from both our previous outpatient study and the current inpatient study indicate that the lack of patient transportation of medical documents was likely due to systemic issues, which could be remedied using various strategies.
- These issues included inconsistent instructions/reminders from doctors (i.e. doctors not asking patients to keep and/or bring back their medical notes) and a lack of formal systems regarding the structure and use of medical documentation. In addition, whilst most patients displayed positive attitudes towards medical documents, HCP reports indicated that some also lacked awareness of their importance for maintaining continuity of care.
- There are a number of ways that patient-held booklets could be used to improve the issues identified. Initially, it would be necessary to include both patients and healthcare staff in the booklet design process to make it context-appropriate and invoke a sense of ownership via collaboration.
- The introduction of the booklet could also be accompanied by supportive training and/or educational material for both HCPs and patients; this would assist in promoting and normalising utilisation.
- Another possibility for encouraging patient retention and/or transportation would be to use incentivisation strategies.
- Overall, given that most patients in the current study reported that they regularly stored and transported medical documents to HCP visits, there is reason to believe that these strategies hold promise.

Some of the above information mentioned above has now been further clarified in the discussion section (see page 30, from line 547) of the manuscript.

2. Discussion, p27, lines 47 and further: Why do the authors expect that using a patient-held booklet would work better than the patient-held documents so far?

Thank you for your question. As mentioned in the response to comment 12 above, our findings indicate that there is a lack of formal systems regarding the structure and use of medical documentation within the study areas. This means that presently, medical documents provided to patients are largely unstructured and often get lost, disposed of or stored and transferred in a disorganised way, meaning that key information is not easily accessible to patients and HCPs alike.

The authors believe that a simple and well-designed patient-held booklet would work better as it could contain structured documents that help to retain more key information and also organise medical documents in a more logical and accessible way. The inclusion of both patients and HCPs in the booklet design process could also help to make it more context-appropriate, as well as invoke a sense of collaborative ownership amongst its users that may increase regular utilisation.

Finally, as well as the findings and insights gained from our previous outpatient and the current inpatient research, patient-held record booklets have also proven to be effective for improving continuity of care and patient satisfaction for outpatients and in maternal and child health across several low, middle and high-income country settings.

For clarification, further detail and references regarding some of the points made above have now been added to the discussion section (see page 30, from line 547) of the manuscript.

Reviewer 1 – Minor comments:

3. Abstract, p2, line 34: abbreviation NCD should be explained first time and abbreviation HPC should be HCP

Thank you for your comment. The abbreviations have now been corrected.

4. In the strengths and limitations it is mentioned that “this is the first qualitative study...”. But a recently published paper of the authors (Humphries et al. Investigating clinical handover and healthcare communication for outpatients with chronic disease in India: A mixed-methods study. PLoSOne. 2018;13(12):e02070511) reports on a comparable topic, although with a focus on outpatients.

Thank you for your comment. This statement required further clarification to indicate that the study is the first, to the best of the authors’ knowledge, to qualitatively explore factors affecting multiple areas of handover communication for chronic disease inpatients in India. This has now been corrected (see page 33, from line 622).

5. Introduction, p5, line 107: please use consistent format for referencing

Thank you for your comment. The reference in question was formatted specifically to indicate that the research is, as of yet, unpublished. However, the latter half of the introduction section (see page 6, from line 133) of the manuscript has now been restructured and published LMIC evidence evidencing a link between quality of discharge planning and patient safety has been utilised to improve clarity.

6. Results: Table 2: Explain abbreviations MPH and MBBS.

Thank you for your comment. All abbreviations have now been explained and a missing footnote (*) has been added that corresponds to the qualifications sections of the table to indicate that HCPs could select more than one answer for that question (see page 17).

7. Data sharing statement: for qualitative studies, the transcripts can be made available, please state whether this is the case or not (and why).

Thank you for your comment. The data sharing statement has now been updated to indicate that the transcripts can be made available from the corresponding author upon reasonable request (see page 38, from line 738).

VERSION 2 – REVIEW

REVIEWER	Dr. Hanneke Merten
----------	--------------------

	Amsterdam UMC, Vrije Universiteit Amsterdam, Department of Public and Occupational Health, Amsterdam Public Health Research Institute
REVIEW RETURNED	04-Oct-2019

GENERAL COMMENTS	The issues from the previous review have been sufficiently addressed and well-explained. The added value of this manuscript is now explicitly included in relation to the outpatient study already published. However, I do feel there is some overlap in both studies, which is probably inevitable with this topic on healthcare communication. Many of the identified themes relate to healthcare organisation in general (such as high patient load, absence of training and digital patient record systems) and the majority of HCPs work both in inpatient and outpatient care, so it seems appropriate that they come up in both the inpatient and outpatient study. On the other hand, I still wonder whether it would have been better to describe them together in one manuscript which would have provided an overall overview of healthcare communication in this setting. Some very minor issues (pages and line numbers refer to tracked changes version of manuscript):  1. page 12, lines 252-253: perhaps include that different patients were interviewed for this inpatient study (instead of just mentioning it was independent)? 2. Page 16, line 363: it mentions 26 HCP transcripts were analysed, but the study included only 21 HCP-interviews? 3. Page 31, line 479, insert 'that' (?) after 'information' 4. Page 35, line 609: remove 'add' after HCP
---

VERSION 2 – AUTHOR RESPONSE

Reviewer 1 – Minor Comments:

1. Page 12, lines 252-253: perhaps include that different patients were interviewed for this inpatient study (instead of just mentioning it was independent)?

Thank you for your comment. The information has now been updated to explain that different patients were interviewed for this study (see page 9, from line 205).

2. Page 16, line 363: it mentions 26 HCP transcripts were analysed, but the study included only 21 HCP-interviews?

Thank you for your query. This was a typo that has now been corrected to clarify that 21 HCP interviews were analysed and included the study (see page 13, line 305).

3. Page 31, line 479, insert 'that' (?) after 'information'

Thank you for your comment. This sentence has now been revised to improve grammar and clarity (see page 31, from line 586).

4. Page 35, line 609: remove 'add' after HCP

Thank you for your comment. This has now been corrected (see page 32, from line 593). The authors have also ensured that the work has again been proof-read and that all grammatical issues have been corrected.